# Asymptotic Behavior of Resolvents of a Convergent Sequence of Convex Functions on Complete Geodesic Spaces

Yasunori Kimura 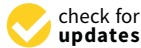 and Keisuke Shindo *

Department of Information Science, Toho University, Miyama, Funabashi, Chiba 274-8510, Japan;
yasunori@is.sci.toho-u.ac.jp
* Correspondence: 7520001s@st.toho-u.jp

**Abstract:** The asymptotic behavior of resolvents of a proper convex lower semicontinuous function is studied in the various settings of spaces. In this paper, we consider the asymptotic behavior of the resolvents of a sequence of functions defined in a complete geodesic space. To obtain the result, we assume the Mosco convergence of the sets of minimizers of these functions.

**Keywords:** asymptotic behavior; geodesic space; convex function; resolvent

## 1. Introduction

The notion of the resolvent for convex functions is one of the most important subjects in the convex minimization problems. We have proposed various resolvents in many spaces and have studied their properties. Moreover, the asymptotic behavior of resolvents at infinity includes crucial problems in studying the properties of resolvents. There are results on the asymptotic behavior of the resolvent of convex functions at infinity. For example, in a Hilbert space $H$, for a proper lower semicontinuous convex function $f\colon H \to ]-\infty, \infty]$, a resolvent $J_f\colon X \to X$ is defined by the following:

$$J_f(x) = \operatorname*{argmin}_{y \in H}\left\{f(y) + \|y - x\|^2\right\}$$

for all $x \in X$. As the asymptotic behavior of this resolvent, the following result is found.

**Theorem 1** (See [1]). *Let $H$ be a Hilbert space and $f\colon H \to ]-\infty, \infty]$ a proper lower semicontinuous convex function. For each $x \in X$, if $\left\{J_{\mu_n f}x\right\}$ is bounded by some sequence $\{\mu_n\} \subset\, ]0, \infty[$ such that $\mu_n \to \infty$, then $\operatorname{argmin} f \neq \varnothing$ and*

$$\lim_{\lambda \to \infty} J_{\lambda f}x = P_{\operatorname{argmin} f}x.$$

On the other hand, geodesic spaces are metric spaces which have some convex structures. In geodesic spaces, many types of resolvents are also proposed and studied. A complete CAT(0) space, which is an example of a geodesic space, is a generalization of Hilbert spaces. In this space, the following resolvent is proposed (see [2]). Let $X$ be a complete CAT(0) space and $f\colon X \to ]-\infty, \infty]$ a proper lower semicontinuous convex function. We define the resolvent $J_f\colon X \to X$ of $f$ by the following equation:

$$J_f(x) = \operatorname*{argmin}_{y \in H}\left\{f(y) + d(y, x)^2\right\}$$

for all $x \in X$. For this resolvent, we can also consider asymptotic behavior at infinity and have results similar to Theorem 1 (see [3]). In these cases, a convex function $f$ is fixed. In a Banach space, the convergence of a sequence for resolvents of maximal monotone operators has been considered in many papers. For example, see [4–10].

Therefore, we will also consider the convergent sequence of convex functions $\{f_n\}$ and their resolvents. We characterize the convergence of a sequence of convex functions by using the set convergence of minimizers. Mosco convergence is one of the useful notions of set convergence. It is defined in Banach spaces and complete admissible CAT($\kappa$) spaces. See [11–13] for more details.

This paper considers the asymptotic behavior of the resolvents of a given convergent sequence of convex functions on a complete CAT(0) space and a complete admissible CAT(1) space. As a convergence of a sequence of convex functions $\{f_n\}$, we suppose that $\{\operatorname{argmin} f_n\}$, the sequence of sets of minimizers of $f_n$, is convergent in the sense of Mosco.

## 2. Preliminaries

Let $X$ be a metric space. For $x, y \in X$, $c_{xy} \colon [0, d(x, y)] \to X$ is called a geodesic with the endpoints $x$ and $y$ if $c_{xy} \colon [0, d(x, y)] \to X$ satisfies the following:

$$\begin{cases} c_{xy}(0) = x; \\ c_{xy}(d(x, y)) = y; \\ d(c_{xy}(u), c_{xy}(v)) = |u - v| \text{ for } u, v \in [0, d(x, y)]. \end{cases}$$

We say that $X$ is a uniquely geodesic space if there exists $c_{xy}$ uniquely for each $x, y \in X$. For $x, y \in X$, a geodesic segment $[x, y]$ joining $x$ and $y$ is an image of $c_{xy}$ defined by $[x, y] = c_{xy}([0, d(x, y)])$. A convex combination $z$ between $x$ and $y$ is a point of $[x, y]$ such that $d(x, z) = (1 - t)d(x, y)$ and $d(z, y) = td(x, y)$, and we denote this $z$ by $tx \oplus (1 - t)y$. Let $X$ be a uniquely geodesic space and $x_1, x_2, x_3 \in X$. A geodesic triangle $\triangle(x_1, x_2, x_3) \subset X$ with vertices $x_1, x_2, x_3$ is defined by $\triangle(x_1, x_2, x_3) = [x_1, x_2] \cup [x_2, x_3] \cup [x_3, x_1]$. For a geodesic triangle $\triangle(x_1, x_2, x_3)$, a comparison triangle $\overline{\triangle}(\overline{x}_1, \overline{x}_2, \overline{x}_3) \subset \mathbb{R}^2$ is defined as a triangle whose vertices $\overline{x}_1, \overline{x}_2, \overline{x}_3$ satisfy $\|\overline{x}_1 - \overline{x}_2\| = d(x_1, x_2)$, $\|\overline{x}_2 - \overline{x}_3\| = d(x_2, x_3)$, $\|\overline{x}_3 - \overline{x}_1\| = d(x_3, x_1)$. Furthermore, for $p \in [x_i, x_j]$ ($i, j = 1, 2, 3$ and $i \neq j$), a comparison point $\overline{p}$ of $p$ is a point on $[\overline{x}_i, \overline{x}_j]$ such that $\|\overline{p} - \overline{x}_i\| = d(p, x_i)$. $X$ is called a CAT(0) space if for any geodesic triangle $\triangle(x_1, x_2, x_3)$, any $p, q \in \triangle(x_1, x_2, x_3)$, and their comparison points $\overline{p}, \overline{q} \in \overline{\triangle}(\overline{x}_1, \overline{x}_2, \overline{x}_3)$, the following holds:

$$d(p, q) \le \|\overline{p} - \overline{q}\|.$$

Let $X$ be a geodesic space and $\triangle(x_1, x_2, x_3)$ a geodesic triangle on $X$. In the same way as above, we define a comparison triangle $\overline{\triangle}(\overline{x}_1, \overline{x}_2, \overline{x}_3) \subset \mathbb{S}^2$. $X$ is called a CAT(1) space if for any geodesic triangle $\triangle(x_1, x_2, x_3)$ with $d(x_1, x_2) + d(x_2, x_3) + d(x_3, x_1) < 2\pi$, any $p, q \in \triangle(x_1, x_2, x_3)$, and their comparison points $\overline{p}, \overline{q} \in \overline{\triangle}(\overline{x}_1, \overline{x}_2, \overline{x}_3)$, it holds that:

$$d(p, q) \le d_{\mathbb{S}^2}(\overline{p}, \overline{q}).$$

An admissible CAT(1) space is a CAT(1) space such that the distance of any two points is smaller than $\pi/2$. Let $X$ be an admissible CAT(1) space and $\{x_n\}$ a sequence of $X$. The sequence $\{x_n\}$ is said to be spherically bounded if there exists $y \in X$ such that $\sup d(x_n, y) < \pi/2$ for all $n \in \mathbb{N}$.

We describe the fundamental properties of complete CAT(0) spaces and complete admissible CAT(1) spaces. The following inequalities are called parallelogram laws.

**Theorem 2** (See [3,14]). *Let $X$ be a complete* CAT(0) *space, $x, y, z \in X$, and $t \in [0, 1]$. Then,*

$$d(tx \oplus (1 - t)y, z)^2 \le td(x, z)^2 + (1 - t)d(y, z)^2 - t(1 - t)d(x, y)^2.$$

**Theorem 3** (See [14]). *Let $X$ be a complete admissible* CAT(1) *space, $x, y, z \in X$, and $t \in [0, 1]$. Then,*

$$\cos d(tx \oplus (1 - t)y, z) \sin d(x, y) \ge \cos d(x, z) \sin td(x, y) + \cos d(y, z) \sin(1 - t)d(x, y).$$

*In particular, for $t = 1/2$, it holds that:*

$$\cos d\left(\frac{1}{2}x \oplus \frac{1}{2}y, z\right) \cos \frac{d(x,y)}{2} \geq \frac{1}{2}\cos d(x,z) + \frac{1}{2}\cos d(y,z),$$

*or equivalently, that:*

$$-\log\cos d\left(\frac{1}{2}x \oplus \frac{1}{2}y, z\right)$$
$$\leq \frac{1}{2}(-\log(\cos d(x,z))) + \frac{1}{2}(-\log(\cos d(y,z)) + \log\cos\frac{d(x,y)}{2}.$$

Let $X$ be a metric space and $\{x_n\}$ a bounded sequence in $X$. For $x \in X$, we assign the following equation:

$$r(x, \{x_n\}) = \limsup_{n\to\infty} d(x, x_n), \quad r(\{x_n\}) = \inf_{x\in X} r(x, \{x_n\}).$$

Then, if $x \in X$ satisfies $r(x, \{x_n\}) = r(\{x_n\})$, it is called an asymptotic center of $\{x_n\}$. Moreover, if for any subsequence of $\{x_n\}$ its asymptotic center is a unique point $x$, we say that $\{x_n\}$ is $\Delta$-convergent to $x$. Any bounded sequences in complete CAT(0) space have a $\Delta$-convergent subsequence. Likewise, any spherically bounded sequences in complete admissible CAT(1) have a $\Delta$-convergent subsequence. See [15–17].

Let $X$ be a complete CAT(0) or complete admissible CAT(1) space, and $C$ a closed convex subset of $X$. Then, for $x \in X$, there exists a unique $x_0 \in C$ such that:

$$d(x_0, x) = \inf_{y\in C} d(y, x).$$

We define $P_C : X \to C$ by the following:

$$P_C(x) = \operatorname*{argmin}_{y\in C} d(y, x).$$

for $x \in X$. This $P_C$ is called a metric projection onto $C$ and has the following properties. If $X$ is a complete CAT(0) space, then

$$d(x, P_C x)^2 + d(P_C x, y)^2 \leq d(x, y)^2$$

for all $y \in C$ and $x \in X$. If $X$ is a complete admissible CAT(1) space, then

$$\cos d(x, P_C x)\cos d(P_C x, y) \geq \cos d(x, y)$$

for all $y \in C$ and $x \in X$.

Let $C_1, C_2, C_3, \ldots$ be nonempty closed convex subsets of a complete CAT(0) or complete admissible CAT(1) space $X$. We define the sets d-Li $C_n$ and $\Delta$-Ls $C_n$ as follows: $v \in$ d-Li $C_n$ if and only if there exists $\{v_n\}$ such that $v_n \to v$ and $v_n \in C_n$ for each $n$; $w \in \Delta$-Ls $C_n$ if and only if there exists a bounded sequence $\{w_i\}$ such that $w \in AC\{w_i\}$ and $w_i \in S_{n_i}$ for each $i$. If a closed convex subset $C_0$ of $X$ satisfies the following:

$$\text{d-Li } C_n \subset C_0 \subset \Delta\text{-Ls } C_n,$$

we say that $\{C_n\}$ converges to $C_0$ in the sense of Mosco and denote M-$\lim_{n\to\infty} C_n = C_0$.

## 3. Main Results

Let $f_1, f_2, f_3, \ldots$ be the proper convex lower semicontinuous functions on a CAT(0) or complete admissible CAT(1) space $X$. As the convergence of a sequence of convex functions $\{f_n\}$, we suppose the following conditions:

(a)  M-$\lim_{n\to\infty} \operatorname{argmin} f_n = \operatorname{argmin} f$;

(b) For all $b \in X$, there exists $\{b_n\}$ such that $b_n \to b$ and $\limsup_{n \to \infty} f_n(b_n) \le f(b)$;

(c) For any subsequence $\{f_{n_i}\}$ of $\{f_n\}$ and a $\Delta$-convergent sequence $\{c_i\}$ whose $\Delta$-limit is $c \in X$, it holds that $f(c) \le \liminf_{i \to \infty} f_{n_i}(c_i)$.

We consider the asymptotic behavior of a resolvent on CAT(0) space. Let $X$ be a complete CAT(0) space, $f \colon X \to \, ]-\infty, \infty]$ a proper convex lower semicontinuous function, and $x \in X$. We say that a function $\varphi \colon [0, \infty[ \, \to \mathbb{R}$ satisfies the condition (A) if the following conditions hold:

- $\varphi$ is increasing;
- $\varphi$ is continuous;
- $\varphi(d(\cdot, x))$ is strictly convex for all $x \in X$;
- $\varphi(t) - kt \to \infty$ as $t \to \infty$, for all constants $k \in \mathbb{R}$.

If $\varphi$ satisfies the condition (A), then the function $f(\cdot) + \varphi(d(\cdot, x))$ has a unique minimizer. We define a resolvent $J_f$ of $f$ with $\varphi$ by the following equation:

$$J_f(x) = \operatorname*{argmin}_{y \in X} \{f(y) + \varphi(d(y, x))\}$$

for $x \in X$. For example, $\varphi_1(t) = t^2$ and $\varphi_2(t) = \tanh t \sinh t$ satisfies these conditions. If we define the resolvent with $\varphi_1(t) = t^2$, it is the resolvent described in the Introduction. For complete CAT($-1$) spaces, which are a special case of CAT(0) spaces, the resolvent with $\varphi_2(t) = \tanh t \sinh t$ is defined and studied in [18].

Now we describe the asymptotic behavior of resolvents for a sequence of convex functions satisfying (a), (b), and (c).

**Theorem 4.** *Let $X$ be a complete* CAT(0) *space, $\{f_n\}$ a sequence of proper convex lower semicontinuous functions from $X$ to $]-\infty, \infty]$, $f$ a proper convex lower semicontinuous function from $X$ to $]-\infty, \infty]$, and $\{\lambda_n\} \subset \, ]0, \infty[$ an increasing sequence diverging to $\infty$. If $\{f_n\}$ and $f$ satisfy the conditions (a), (b), and (c), then for $x \in X$, we have:*

$$\lim_{n \to \infty} J_{\lambda_n f_n} x = P_{\operatorname{argmin} f} x.$$

**Proof.** Let $x \in X$. We put $x_n = J_{\lambda_n f_n} x$ and $p = P_{\operatorname{argmin} f} x$. Since $p \in \operatorname{argmin} f_0 \subset$ d-Li $\operatorname{argmin} f_n$ from the condition (a), there exists $\{a_n\}$ such that $a_n \in \operatorname{argmin} f_n$ for each $n$ and $a_n \to p$. Since points $a_n$ and $x_n$ are minimizers of $f_n$ and $f_n(\cdot) + \varphi(d(\cdot, x))$, respectively, then we have the follwing equation:

$$f_n(a_n) + \frac{1}{\lambda_n} \varphi(d(x_n, x)) \le f_n(x_n) + \frac{1}{\lambda_n} \varphi(d(x_n, x))$$
$$\le f_n(a_n) + \frac{1}{\lambda_n} \varphi(d(a_n, x)).$$

Thus, we get $\varphi(d(x_n, x)) \le \varphi(d(a_n, x))$, which is equivalent to $d(x_n, x) \le d(a_n, x)$. Since $\{a_n\}$ is a convergent sequence, and $\{a_n\}$ is bounded, this implies that $\{x_n\}$ is also bounded. Take a subsequence $\{x_{n_i}\}$ of $\{x_n\}$ arbitrarily. There exists a $\Delta$-convergent subsequence $\{x_{n_{ij}}\}$ of $\{x_{n_i}\}$ to some $q \in X$. From the condition (b), there exists $\{b_n\}$ such that $b_n \to p$ and $\limsup_{n \to \infty} f_n(b_n) \le f(p)$. Furthermore, using the condition (c), we get $\liminf_{j \to \infty} f_{n_{ij}}(x_{n_{ij}}) \le f(q)$ as $x_{n_{ij}} \overset{\Delta}{\to} q$. From the definition of the resolvent, we have the following equation:

$$f_{n_{ij}}(x_{n_{ij}}) + \frac{1}{\lambda_{n_{ij}}} \varphi(d(x_{n_{ij}}, x)) \le f_{n_{ij}}(b_{n_{ij}}) + \frac{1}{\lambda_{n_{ij}}} \varphi(d(b_{n_{ij}}, x)).$$

By the boundedness of $\{d(x_n, x)\}$ and $\{d(a_n, x)\}$, letting $j \to \infty$, we have the following:

$$f(q) \leq \liminf_{j \to \infty} f_{n_{ij}}(x_{n_{ij}}) \leq \liminf_{j \to \infty} f_{n_{ij}}(b_{n_{ij}}) \leq \limsup_{j \to \infty} f_{n_{ij}}(b_{n_{ij}}) \leq f(p).$$

This implies that $q \in \operatorname{argmin} f$. Since $d(x_{n_{ij}}, x) \leq d(a_{n_{ij}}, x)$, we let $j \to \infty$ again and get the following:

$$d(p, x) \leq d(q, x) \leq \liminf_{j \to \infty} d(x_{n_{ij}}, x) \leq \limsup_{j \to \infty} d(x_{n_{ij}}, x) \leq \lim_{j \to \infty} d(a_{n_{ij}}, x) = d(p, x).$$

Hence, we have $q = p$ and $d(x_{n_{ij}}, x) \to d(p, x)$. Since $a_{n_{ij}}$ is a minimizer of $f_{n_{ij}}$ and $f_{n_{ij}}$ is convex, we have the following equations:

$$
\begin{aligned}
& f_{n_{ij}}(x_{n_{ij}}) + \frac{1}{\lambda_{n_{ij}}} \varphi(d(x_{n_{ij}}, x)) \\
&\leq f_{n_{ij}}\left(\frac{x_{n_{ij}} \oplus a_{n_{ij}}}{2}\right) + \frac{1}{\lambda_{n_{ij}}} \varphi\left(d\left(\frac{x_{n_{ij}} \oplus a_{n_{ij}}}{2}, x\right)\right) \\
&\leq \frac{f_{n_{ij}}(x_{n_{ij}}) + f_{n_{ij}}(a_{n_{ij}})}{2} + \frac{1}{\lambda_{n_{ij}}} \varphi\left(d\left(\frac{x_{n_{ij}} \oplus a_{n_{ij}}}{2}, x\right)\right) \\
&\leq f_{n_{ij}}(x_{n_{ij}}) + \frac{1}{\lambda_{n_{ij}}} \varphi\left(d\left(\frac{x_{n_{ij}} \oplus a_{n_{ij}}}{2}, x\right)\right),
\end{aligned}
$$

and hence,

$$d(x_{n_{ij}}, x) \leq d\left(\frac{x_{n_{ij}} \oplus a_{n_{ij}}}{2}, x\right).$$

From the parallelogram law of CAT(0) space, we get the following:

$$d(x_{n_{ij}}, x)^2 \leq d\left(\frac{x_{n_{ij}} \oplus a_{n_{ij}}}{2}, x\right)^2 \leq \frac{1}{2} d(x_{n_{ij}}, x)^2 + \frac{1}{2} d(a_{n_{ij}}, x)^2 - \frac{1}{4} d(x_{n_{ij}}, a_{n_{ij}})^2.$$

Since both $\{d(a_{n_{ij}}, x)\}$ and $\{d(x_{n_{ij}}, x)\}$ are convergent to $d(p, x)$, we have:

$$d(x_{n_{ij}}, a_{n_{ij}})^2 \leq 2(d(a_{n_{ij}}, x)^2 - d(x_{n_{ij}}, x)^2) \to 0$$

which implies that $x_{n_{ij}} \to p$. Then, any subsequence $\{x_{n_i}\}$ of $\{x_n\}$ has a convergent subsequence $\{x_{n_{ij}}\}$, which tends to $p$. From these facts, we get a desired result. $\square$

From this theorem, we have the following corollaries. Suppose $f_n = f$ for all $n \in \mathbb{N}$. Then $\{f_n\}$ obviously satisfies the conditions (a), (b), and (c).

**Corollary 1.** *Let $X$ be a complete* CAT(0) *space, $f$ a proper convex lower semicontinuous function from $X$ to $]-\infty, \infty]$, and $\varphi \colon [0, \infty[ \to \mathbb{R}$ a function satisfying the condition (A). For a positive real number $\lambda$, define $J_{\lambda f} \colon X \to X$ by the following equation:*

$$J_{\lambda f}(a) = \operatorname*{argmin}_{y \in X} \{\lambda f(y) + \varphi(d(y, a))\}$$

*for $a \in X$. Then, for each $x \in X$,*

$$\lim_{\lambda \to \infty} J_{\lambda f} x = P_{\operatorname{argmin} f} x.$$

Let $\{C_n\}$ be a sequence of nonempty closed convex subsets which converges to $C$ in the sense of Mosco. If $\{f_n\} = \{i_{C_n}\}$ and $f = i_C$, then $\operatorname{argmin} f_n = C_n$ and $\operatorname{argmin} f = C$,

where $i_C$ is the indicator function of $C$. Since $\{C_n\}$ converges to $C$, $\{i_{C_n}\}$ and $i_C$ satisfy the condition (a). They also satisfy the conditions (b) and (c).

**Corollary 2.** *Let $X$ be a complete* CAT(0) *space, $\{C_n\}$ a sequence of nonempty closed convex subsets of $X$, and $C$ a nonempty closed convex subset of $X$. If $\{C_n\}$ converges to $C$ in the sense of Mosco, then for each $x \in X$,*

$$\lim_{n \to \infty} P_{C_n} x = P_C x.$$

Similarly, we consider asymptotic behavior of a resolvent on CAT(1) space. Let $X$ be a complete admissible CAT(1) space. We say $\varphi \colon [0, \pi/2[ \to \mathbb{R}$ satisfies the condition (B) if the following hold:

- $\varphi$ is increasing;
- $\varphi$ is continuous;
- $\varphi(d(\cdot, x))$ is strictly convex for all $x \in X$;
- $\varphi(t) \to \infty$ as $t \to \pi/2$.

Then, the set $\operatorname{argmin}_{y \in X} \{f(y) + \varphi(d(y, x))\}$ is a singleton for all $x \in X$, and we define $J_f$ in a similar way. For example, $\varphi_3(t) = \tan t \sin t$ and $\varphi_4(t) = -\log \cos t$ satisfy the conditions above. On complete admissible CAT(1) spaces, resolvents by using these functions are defined and their properties are studied in [19,20].

We consider that the asymptotic behavior of resolvents for a sequence of convex functions satisfies (a), (b), and (c).

**Theorem 5.** *Let $X$ be a complete admissible* CAT(1) *space, $\{f_n\}$ a sequence of proper convex lower semicontinuous functions from $X$ to $]-\infty, \infty]$, $f$ a proper convex lower semicontinuous function from $X$ to $]-\infty, \infty]$, and $\{\lambda_n\} \subset ]0, \infty[$ an increasing sequence diverging to $\infty$. If $\{f_n\}$ and $f$ satisfy the conditions (a), (b), and (c), then for $x \in X$,*

$$\lim_{n \to \infty} J_{\lambda_n f_n} x = P_{\operatorname{argmin} f} x.$$

**Proof.** In the same way as the proof of Theorem 5, if we take $\{x_{n_{ij}}\}$ with the same procedure, it hold that $d(x_{n_{ij}}, x) \to d(p, x)$ and

$$d(x_{n_{ij}}, x) \le d\left(\frac{x_{n_{ij}} \oplus a_{n_{ij}}}{2}, x\right).$$

From the parallelogram law of CAT(1) space, we get the following equations:

$$
\begin{aligned}
-\log(\cos d(x_{n_{ij}}, x)) &\le -\log\left(\cos d\left(\frac{x_{n_{ij}} \oplus a_{n_{ij}}}{2}, x\right)\right) \\
&\le \frac{1}{2}(-\log(\cos d(x_{n_{ij}}, x))) + \frac{1}{2}(-\log(\cos d(a_{n_{ij}}, x))) \\
&\quad - \frac{1}{4}(-\log(\cos d(x_{n_{ij}}, a_{n_{ij}}))).
\end{aligned}
$$

Then, we have

$$-\log(\cos d(x_{n_{ij}}, a_{n_{ij}})) \le 2(-\log(\cos d(x_{n_{ij}}, x)) - \log(\cos d(a_{n_{ij}}, x))).$$

This implies that $x_{n_{ij}} \to p$, and we get $J_{\lambda_n f_n} x \to P_{\operatorname{argmin} f} x.$　□

As well as for the case of CAT(0) spaces, we obtain the following corollaries in CAT(1) spaces.

**Corollary 3.** *Let $X$ be a complete admissible* CAT(1) *space, $f$ a proper convex lower semicontinuous function from $X$ to $]-\infty, \infty]$, and $\varphi\colon [0, \pi/2[ \to \mathbb{R}$ a function satisfying the condition (B). For a positive real number $\lambda$, define $J_{\lambda f}\colon X \to X$ by the following equation:*

$$J_{\lambda f}(a) = \operatorname*{argmin}_{y \in X}\{\lambda f(y) + \varphi(d(y,a))\}$$

*for $a \in X$. Then, for each $x \in X$, we have:*

$$\lim_{\lambda \to \infty} J_{\lambda f} x = P_{\operatorname{argmin} f} x.$$

**Corollary 4.** *Let $X$ be a complete admissible* CAT(1) *space, $\{C_n\}$ a sequence of nonempty closed convex subsets of $X$, and $C$ a nonempty closed convex subset of $X$. If $\{C_n\}$ converges to $C$ in the sense of Mosco, then for each $x \in X$, we have:*

$$\lim_{n \to \infty} P_{C_n} x = P_C x.$$

## 4. Applications to Hilbert Spaces

Finally, we consider the applications of our results to the case of a Hilbert space. Because the class of complete CAT(0) spaces includes that of Hilbert spaces, we can get some results in Hilbert spaces directly. The definitions of conditions for functions $\varphi$ and $\{f_n\}$ are applied to those in CAT(0) spaces.

**Theorem 6.** *Let $H$ be a Hilbert space, $\{f_n\}$ a sequence of proper convex lower semicontinuous functions from $X$ to $]-\infty, \infty]$, $f$ a proper convex lower semicontinuous function from $X$ to $]-\infty, \infty]$, and $\{\lambda_n\} \subset ]0, \infty[$ an increasing sequence diverging to $\infty$. Suppose $\{f_n\}$ and $f$ satisfy the conditions (a), (b), and (c), and $\varphi\colon [0, \infty[ \to \mathbb{R}$ satisfies the condition (A). Define $J_{\lambda_n f_n}\colon X \to X$ by the following:*

$$J_{\lambda_n f_n}(a) = \operatorname*{argmin}_{y \in X}\{\lambda_n f_n(y) + \varphi(\|y - a\|)\}$$

*for $a \in X$. Then, for each $x \in X$*

$$\lim_{n \to \infty} J_{\lambda_n f_n} x = P_{\operatorname{argmin} f} x.$$

Using this result, we can get following famous theorems. First, if we consider the case that a convex function is fixed and $\varphi(t) = t^2$, we can get Theorem 1. Next, considering the case that convex functions are the indicator functions of some convex sets, we obtain the following theorem.

**Theorem 7** (See [21]). *Let $H$ be a Hilbert space, $\{C_n\}$ a sequence of nonempty closed convex subsets of $X$, and $C$ a nonempty closed convex subset of $X$. If $\{C_n\}$ converges to $C$ in the sense of Mosco, then for each $x \in X$,*

$$\lim_{n \to \infty} P_{C_n} x = P_C x.$$

In conclusion, we summarize the results in this paper. For a given sequence $\{f_n\}$ of proper lower semicontinuous functions converging to $f$ in the sense of the conditions (a), (b), and (c), we consider the corresponding sequence of resolvents $\{J_{\lambda_n f_n}\}$ with a positive real sequence $\{\lambda_n\}$ diverging to $\infty$. The main results imply the pointwise convergence of this sequence to the metric projection onto $\operatorname{argmin} f$ in the setting of a CAT(0) and a CAT(1) space, respectively. We can apply them to the asymptotic behavior of the resolvent for a single function at $\infty$, and a convergence theorem for a sequence of metric projections.

**Author Contributions:** The authors (Y.K. and K.S.) have contributed to this work on an equal basis. All authors have read and agreed to the published version of the manuscript.

**Funding:** This work was partially supported by JSPS KAKENHI, Grant Number JP21K03316.

**Conflicts of Interest:** The authors declare no conflict of interest.

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
