# Peer review of "Asymptotic Behavior of Resolvents of a Convergent Sequence of Convex Functions on Complete Geodesic Spaces"

_axioms, doi:10.3390/axioms11010021_

Round 1

Reviewer 1 Report

In this paper is considered X a metric space. If, for every points x, y from X, we can define a geodesic cxy (see p. 2), then X  becomes a geodesic space. The spaces CAT(0) and CAT(1) are two types of geodesic spaces, which are used in this paper.  If  f : X→ ]-∞,+ ∞] is a proper convex lower semicontinuous function, a resolvent  Jf : X X is defined (see p. 1).  The authors study asymptotic behavior of resolvents of a sequence of functions in complete spaces. CAT(0) and complete admissible CAT(1) spaces, by assuming  Mosco convergence of the sets of minimizers of these functions.

          The paper is carelessly written. There are not examples for the notions used in this article and comparison with other results from this field of research. To give the motivation for this study are necessary to present or to quote applications of the new results. Many papers from References are not quoted in this article and there are results (e. g. Theorems 2.1 and 2.2) and statements without references.

I notice several misprints:

-  p. 1, line 2 of Theorem 1.1;

-  p. 1, line 7 below,

-  p. 2, line 14, (the definition of a comparison triangle),

-  p. 3, line 3, the third term.

Author Response

Dear the reviewer, 

Thank you for your variable comments.

Following your suggestions, we revised our manuscript.  The details are as follows:

  1. We explained the reasons why a generalized function \phi was invoked, and why the conditions (A) or (B) are assumed; see p4 and p6.
  2. We revised the introduction and clarified the history and the motivation of our work.
  3. We added Section 4 and provide some examples in the case of Hilbert spaces.
  4. We checked the reference and refer all the items in the reference from the manuscript. 
  5. We corrected typos mentioned in the review.

Reviewer 2 Report

attached

Author Response

Dear the reviewer, 

Thank you for your variable comments.

Following your suggestions, we revised our manuscript.  The details are as follows:

  1. We explained the reasons why a generalized function \phi was invoked, and why the conditions (A) or (B) are assumed; see p4 and p6.
  2. We revised the introduction and clarified the history and the motivation of our work.
  3. We added Section 4 and provide some examples in the case of Hilbert spaces.
  4. We checked the reference and refer all the items in the reference from the manuscript. 
  5. We corrected minor misprints.

Reviewer 3 Report

The authors have studied the asymptotic behavior of the resolvents of some convergence sequence of functions. They investigate this behavior in spaces that are completely geodesic. The main theorem is proved using the Mosco convergence of sets of certain minimizers of considered functions.

The paper is well structured and the main theorems offer new results about the topic. The results have a form of certain conditions for which these sets of functions are convergent.

On page 4 (row +3) there isn't a capital letter in the second  sentence. I would also appreciate some conclusion at the end of the article, in this form it looks a bit incomplete, which it is not. One sentence would suffice.

Based on what's mentioned above and because the results are new I do recommend the paper for publication.

Author Response

Dear the reviewer, 

Thank you for your variable comments.

Following your suggestions, we revised our manuscript.  The details are as follows:

  1. In the end of the manuscript, we added the conclusion of this paper.
  2. We explained the reasons why a generalized function \phi was invoked, and why the conditions (A) or (B) are assumed; see p4 and p6.
  3. We revised the introduction and clarified the history and the motivation of our work.
  4. We added Section 4 and provide some examples in the case of Hilbert spaces.
  5. We checked the reference and refer all the items in the reference from the manuscript. 
  6. We corrected typos mentioned in the review.

Round 2

Reviewer 1 Report

I have not new remarks.